# Erosion–Corrosion Behavior of 90/10 and 70/30 Copper–Nickel Tubes in 1 wt% NaCl Solution

Lei Wu [1,2], Aili Ma [1,*], Lianmin Zhang [1], Guangming Li [3], Lingyue Hu [3], Zhengbin Wang [1] and Yugui Zheng [1,*]

[1] CAS Key Laboratory of Nuclear Materials and Safety Assessment, Institute of Metal Research, Chinese Academy of Sciences, 62 Wencui Road, Shenyang 110016, China
[2] School of Materials Science and Engineering, University of Science and Technology of China, 72 Wenhua Road, Shenyang 110016, China
[3] Wuhan Second Ship Design and Research Institute, 450 Zhongshan Road, Wuhan 430064, China
[*] Correspondence: alma@imr.ac.cn (A.M.); ygzheng@imr.ac.cn (Y.Z.)

**Abstract:** The erosion–corrosion behavior of 90/10 and 70/30 copper–nickel tubes was investigated by in situ electrochemical tests on a self-built loop apparatus and ex situ surface characterization. The corrosion product film that formed at 1.5 m/s for the 90/10 tube and at 0.5 m/s for the 70/30 tube showed the best corrosion resistance. For the 90/10 tube, a continuous film existed below 3 m/s and mainly inhibited a cathodic reaction. For the 70/30 tube, a continuous film existed in the range of 0.5–4.7 m/s and was more similar to typical passive film electrochemically, although it was cracked at 4–4.7 m/s. So, the "critical flow velocity" of the 90/10 tube was between 3 m/s and 4 m/s, and that of the 70/30 tube was beyond 4.7 m/s.

**Keywords:** erosion; corrosion; critical flow velocity; copper-nickel alloy; film





## 1. Introduction

Copper–nickel alloy tubes are mainly used in flowing seawater environments, such as heat exchanging systems in vessels and coastal power plants, seawater piping systems in offshore platforms, and multistage flash systems in seawater desalination factories, which is due to their good thermal conductivity, formability, antifouling, and corrosion-resistant properties [1,2]. Among them, the amount of 90/10 copper–nickel alloy is the largest, and the 70/30 copper–nickel alloy is used in the critical parts with more demanding conditions. The excellent corrosion resistance of copper–nickel alloys mainly depends on the corrosion product film formed on the surface, which reduces the diffusion rate of ions and, thereby, the corrosion rate [3]. The corrosion product film of copper–nickel alloys formed in stagnant seawater is mainly composed of a compact and defective $Cu_2O$ inner layer doped with $Ni^+$ and a porous $(Cu, M)_2(OH)_3Cl$ (M is Ni, Fe, Mn) outer layer [4,5]. Under flowing conditions, F.P. Ijsseling mentioned that the well-formed corrosion product film could resist the erosion–corrosion of fluid effectively and play a protective role on the substrate metal [6]. However, premature failures of copper–nickel alloy tubes still happen around the world, which are mainly attributed to the degradation of the corrosion product films in flowing fluid [7–9]. This shows that our understanding of the formation and evolution of the corrosion product film of copper–nickel alloys in a flowing environment is not enough.

There exists the critical flow velocity ($v_{crit}$) phenomenon for the corrosion behavior of copper alloys in flowing seawater; that is, when the flow velocity is lower than the $v_{crit}$, the corrosion rate of copper alloys is low, and when the flow velocity exceeds the $v_{crit}$, the corrosion rate increases sharply [10–12]. At present, the $v_{crit}$ values of copper–nickel alloys are controversial. K.D. Efird studied the critical velocities of several kinds of copper alloy sheets using a loop erosion-corrosion apparatus and found that the $v_{crit}$ of 90/10 and 70/30 copper–nickel sheets in seawater are 4.5 m/s and 4.1 m/s, respectively [13]. S. Henrikson et al., studied the critical flow velocity of copper alloy tubes and concluded

that the $v_{crit}$ of 90/10 and 70/30 copper–nickel tubes are less than 3.5 m/s and larger than 4.5 m/s, respectively [14]. Additionally, some researchers suggested that the $v_{crit}$ of 90/10 copper–nickel was about 3 m/s [15,16]. The controversy regarding the critical flow velocity of copper–nickel alloy in seawater is related to the characteristics of its corrosion product film. Different from the rapid film-forming ability of stainless steel and other passivation systems, the corrosion product film of copper–nickel alloys takes a long time (days to dozens of days) to form [2]. It can be speculated that different states of corrosion product film, when tested, will lead to different critical flow velocities measured. The controversy regarding the critical flow velocity of copper–nickel alloys is also related to the uncertainty about the $v_{crit}$ mechanism model for copper alloys. The current mechanism model is based on that of the passivation system (such as stainless steels), which is based on the relative magnitude of the wall shear stress ($\tau_w$) generated by the fluid and the bonding force between the corrosive product film and the substrate. When $v_{crit}$ is reached and $\tau_w$ exceeds the film/substrate interface bonding force, the whole double-layer film is peeled off, causing the corrosion rate to rise sharply, and the critical shear stress calculated for the 90/10 copper–nickel alloy is 43.1 N/m² [13]. However, Bianchi et al., pointed out that the wall shear stress under service conditions is far from enough to directly peel off the inner $Cu_2O$ film, so a mixed physical and diffusion-controlled dissolution model is proposed [17]. Here, when the flow velocity exceeds the $v_{crit}$, the outer layer is removed mechanically, and consequently, the dissolution of the exposed $Cu_2O$ film is accelerated. Its thickness is reduced, leading to the substrate being exposed and the corrosion accelerating [18]. It can be seen that the controversial point about the critical flow velocity mechanism model of copper alloys lies in whether the shear force of the critical flow velocity removes the whole double-layer film or just the outer layer of the film. However, this issue cannot be clarified by the published literature.

In this study, the erosion–corrosion behavior of 90/10 and 70/30 copper–nickel tubes in 1 wt% NaCl solution was studied using a self-built loop apparatus, which can more effectively simulate the hydrodynamic regime that occurs in the industrial environment when compared with the rotating cylinder/disk electrodes [19,20] and the impingement jet apparatus [11,21]. Both the in situ electrochemical monitoring and the ex situ corrosion product film analysis were carried out. The purpose of this work is to determine the $v_{crit}$ values of the 90/10 and 70/30 copper–nickel tubes, to clarify the relationship between $v_{crit}$ and the evolution of corrosion product film, and also to enrich the critical flow velocity mechanism of copper–nickel alloys.

## 2. Material and Methods

The experimental materials were 90/10 and 70/30 copper–nickel alloy tubes with a specification of Φ25 mm × 2 mm. The composition of the tubes is displayed in Table 1. The metallography on the cross-section of the tubes is displayed in Figure 1. Both the annealed 90/10 copper–nickel alloy and the annealed 70/30 copper–nickel alloy have equiaxed grains and twins.

**Table 1.** Compositions (wt%) of the experimental copper–nickel tubes.

| Materials | Ni | Fe | Mn | C | Pb | S | P | Cu |
|---|---|---|---|---|---|---|---|---|
| 90/10 copper–nickel tube | 10.72 | 1.68 | 0.76 | 0.0024 | 0.002 | 0.0025 | 0.003 | Bal. |
| 70/30 copper–nickel tube | 30.4 | 0.75 | 0.88 | 0.04 | 0.001 | 0.002 | 0.004 | Bal. |

Samples were divided into corrosion morphology samples and electrochemical samples. The former samples were short tubes cut from the as-received tube, and the latter samples were saddle-like pieces cut from the as-received tubes with a projected area of 1.0 cm². Additionally, nylon fixtures were used to hold the electrochemical samples. A detailed description of the samples and their preparation progress can be found in our previous research [22].

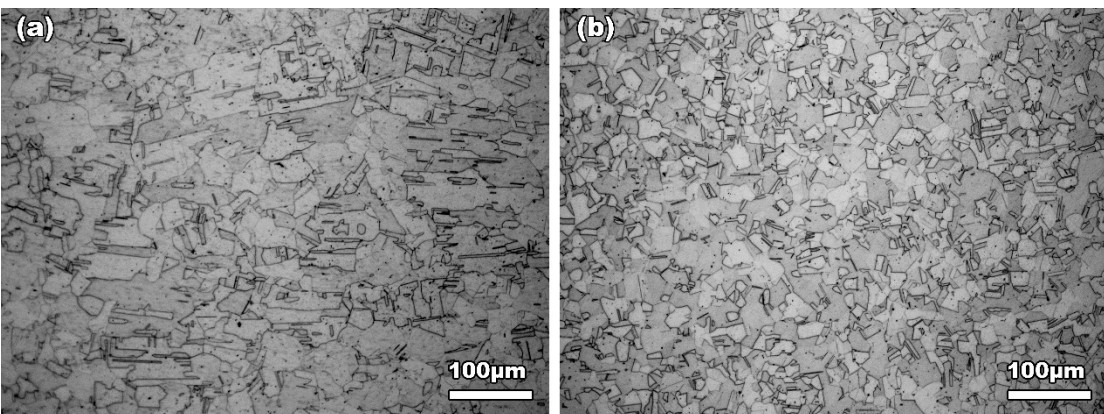

**Figure 1.** Metallographic structure of the (**a**) 90/10 and (**b**) 70/30 copper–nickel tubes used in this work.

To implement the erosion–corrosion test, a self-built loop apparatus was established, which consists of the testing loop and the cooling loop. The cooling loop was used to control the temperature of the solution. Additionally, the main body of the testing loop was assembled from 316 stainless steel tubes with the same dimension as the copper–nickel tubes used in this work, and their inner walls were sprayed with a special anti-corrosion coating. In particular, a "five-way pipe" was used as the electrochemical test section, which was installed in the testing loop and held the three-electrode system. Moreover, the three-electrode system was connected to a Gamry Reference 1000 workstation (Gamry Instruments, Inc., Philadelphia, PA, USA) for in situ electrochemical monitoring. The details of the loop apparatus and the electrochemical test section, as well as the corrosion morphology sample section, can also be found in our previous research [22].

A 1 wt% NaCl solution was prepared by using analytical grade sodium chloride and distilled water to be used as a test solution to simulate brackish water in the coastal region. During the 10-day erosion–corrosion test, 0.5 m/s, 1.5 m/s, 3 m/s, 4 m/s, and 4.7 m/s were chosen as the testing flow velocities. Electrochemical impedance spectroscopy (EIS) was carried out over the frequency range of 100 kHz~10 mHz with a sine amplitude perturbation of $\pm 10$ mV (vs. OCP) after 30 min of open circuit potential (OCP) monitoring. After the EIS test on the last day, the potentiodynamic polarization was carried out in the polarization range of $-0.5$ V–1.2 V (vs. OCP) with a scan rate of 0.3333 mV/s.

Surface and cross-section morphology and elemental composition of the 90/10 and 70/30 copper–nickel samples after the 10-day erosion–corrosion test were obtained by an FEI XL30 field emission gun environment scanning electron microscope and energy dispersive X-ray detector (FEG-ESEM/EDX). Surface morphology samples were cut from the corroded short tubes in small pieces. Cross-section morphology samples were the corroded short tubes mounted in the conductive resin and mechanically ground to 5000# with a series of silicon carbide abrasive papers, then polished with 1.0 μm diamond paste. For better conductivity, the samples were sprayed with carbon.

The XPS measurement was performed on the inner wall of small pieces cut from the corroded short tubes. Because the film is too thick to be sputtered through by argon ions at an affordable experimental cost, two samples were prepared for the surface analysis and the middle analysis, respectively. The sample for the surface analysis is in its original state after the 10-day erosion–corrosion test; the sample for the middle analysis is scratched by a blade until the loose corrosion product layer on the surface is removed. A similar method was used in our previous work [5,23]. The XPS measurement was performed using a VG ESCALAB 250 X-ray photoelectron spectrometer. The photoelectrons were excited with an Al Ka (1486.6 eV) X-ray source, and the analyzer pass energy was 50.0 eV. Binding energies (BE) were calibrated against the surface carbon contamination at 284.6 eV. During the test, signals were collected after 10 s of sputtering to exclude possible contaminants adsorbing

on the surface. Sputtering was conducted using an argon ion beam with an energy level of 2.0 keV, a target current of 2.0 $\mu$A/cm$^2$, and a pressure of $7.7 \times 10^{-9}$ MPa. The sputtering rate was estimated to be about 0.2 nm/s (vs. $Ta_2O_5$).

## 3. Results

### 3.1. In Situ Electrochemical Monitoring during Erosion–Corrosion Test

3.1.1. Electrochemical Impedance Spectrum

The EIS results and fitting plots for 90/10 and 70/30 samples during the 10-day erosion–corrosion test at different flow velocities are shown in Figures 2 and 3, respectively, and the corresponding fitting electrochemical parameters are summarized in Tables 2 and 3. The fitting quality is judged by the chi-square ($\chi^2$) value shown in Tables 2 and 3, which demonstrates that the most fitting results are of good quality. As can be seen in the Nyquist plots in Figures 2 and 3, 3 capacitance arcs are displayed for the 90/10 and 70/30 samples tested at 0.5 m/s and 1.5 m/s; in contrast, two capacitance arcs are displayed for the 90/10 and 70/30 samples tested at 3 m/s, 4 m/s, and 4.7 m/s. Therefore, equivalent circuit 1 in Figures 2f1 and 3f1 is adopted to fit the EIS data of 90/10 and 70/30 samples tested at the lower flow velocities of 0.5 m/s and 1.5 m/s. Equivalent circuit 2 in Figures 2f2 and 3f2 is adopted to fit the EIS data of the 90/10 and 70/30 samples tested at the higher flow velocities of 3 m/s, 4 m/s, and 4.7 m/s. In the equivalent circuits, $R_s$ is the solution resistance, $R_{ct}$ is the charge transfer resistance, CPE1 is the constant phase element parallel to $R_{ct}$, $R_{f1}$ is the outer layer resistance or the resistance of the single-layer corrosion product film, CPE2 is the constant phase element parallel to $R_{f1}$, $R_{f2}$ is the inner layer resistance, and CPE3 is the constant phase element parallel to $R_{f2}$. It should be noted that a constant phase element (CPE) is utilized to replace the ideal capacitance due to the heterogeneous surface.

As shown in Figure 2a2–e2 and Figure 3a2–e2, the peak phase angle in the phase angle plots of the 90/10 samples ranges between 40° and 60°, which demonstrates no typical passive films formed on the 90/10 samples [24]. The peak phase angle of the 70/30 samples ranges between 60° and 80°, indicating the better compactness of the corrosion product film formed on the 70/30 samples. Since the radius of the capacitance arc in Nyquist plots can reflect the corrosion resistance, as shown in Figure 2a1–e1, the corrosion resistance of the 90/10 sample at each flow velocity generally increases with erosion–corrosion time. For the 70/30 copper–nickel sample, as shown in Figure 3a1–e1, the radius of the capacitance arc generally increases with erosion–corrosion time at all flow velocities tested except for 4 m/s, where the capacitance arc radius fluctuates significantly during the erosion–corrosion test.

The film resistance ($R_f = R_{f1} + R_{f2}$) collected from Tables 2 and 3, which reflects the protectiveness of the corrosion product film formed on the 90/10 and 70/30 samples, is displayed in Figure 4. As for the 90/10 sample, the film resistance increases rapidly with time during the first 12 h of the erosion–corrosion test at different flow velocities, but the increasing rates at lower flow velocities are higher than those at higher flow velocities, then the increasing rates at all the flow velocities tested slow down in the following days, which may be attributed to the fast film formation process in the initial stage and the slow film stabilization process during the following days [6,25]. As for the 70/30 sample, a similar trend of the film resistance evolution with time is obtained at different flow velocities except for at 4 m/s, where $R_f$ fluctuates with time during the 10-day erosion–corrosion test. After the 10-day erosion–corrosion, the ranking of film resistance ($R_f$) is $R_{f\,1.5\,m/s} > R_{f\,3\,m/s} > R_{f\,0.5\,m/s} > R_{f\,4\,m/s} > R_{f\,4.7\,m/s}$ for the 90/10 sample and is $R_{f\,0.5\,m/s} > R_{f\,3\,m/s} > R_{f\,1.5\,m/s} > R_{f\,4\,m/s} \approx R_{f\,4.7\,m/s}$ for the 70/30 sample. Overall, the $R_f$ of the 70/30 samples is higher than that of the 90/10 sample at the flow velocities tested, which suggests that the 70/30 sample exhibits better corrosion resistance than the 90/10 sample.

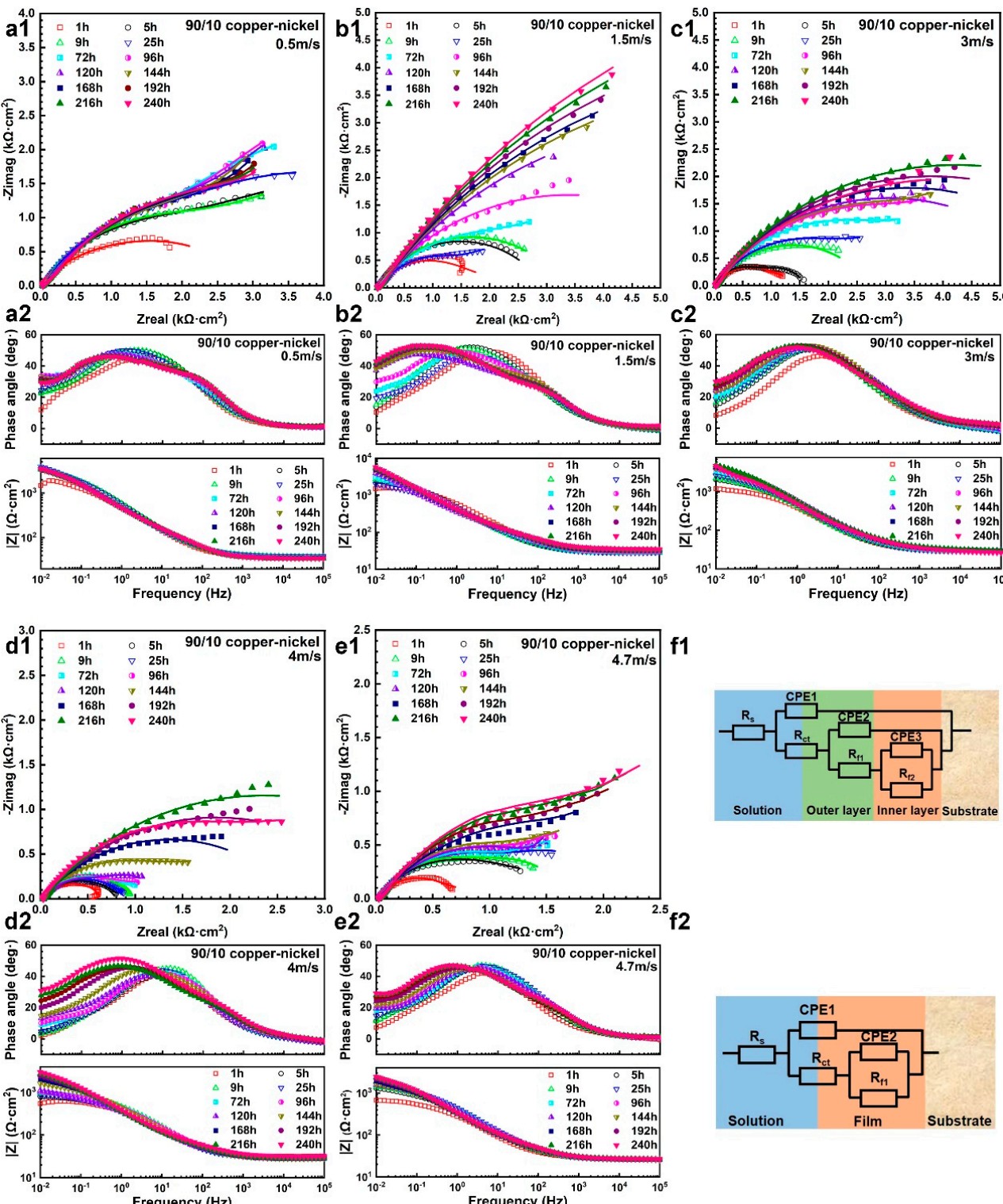

**Figure 2.** Nyquist plots and Bode plots of the 90/10 samples monitored during the 10-day erosion–corrosion tests at flow velocities of 0.5 m/s (**a1**,**a2**), 1.5 m/s (**b1**,**b2**), 3 m/s (**c1**,**c2**), 4 m/s (**d1**,**d2**), and 4.7 m/s (**e1**,**e2**); the equivalent circuits used for EIS fitting tested at lower flow velocities of 0.5 m/s and 1.5 m/s (**f1**) and at higher flow velocities of 3 m/s, 4 m/s, and 4.7 m/s (**f2**).

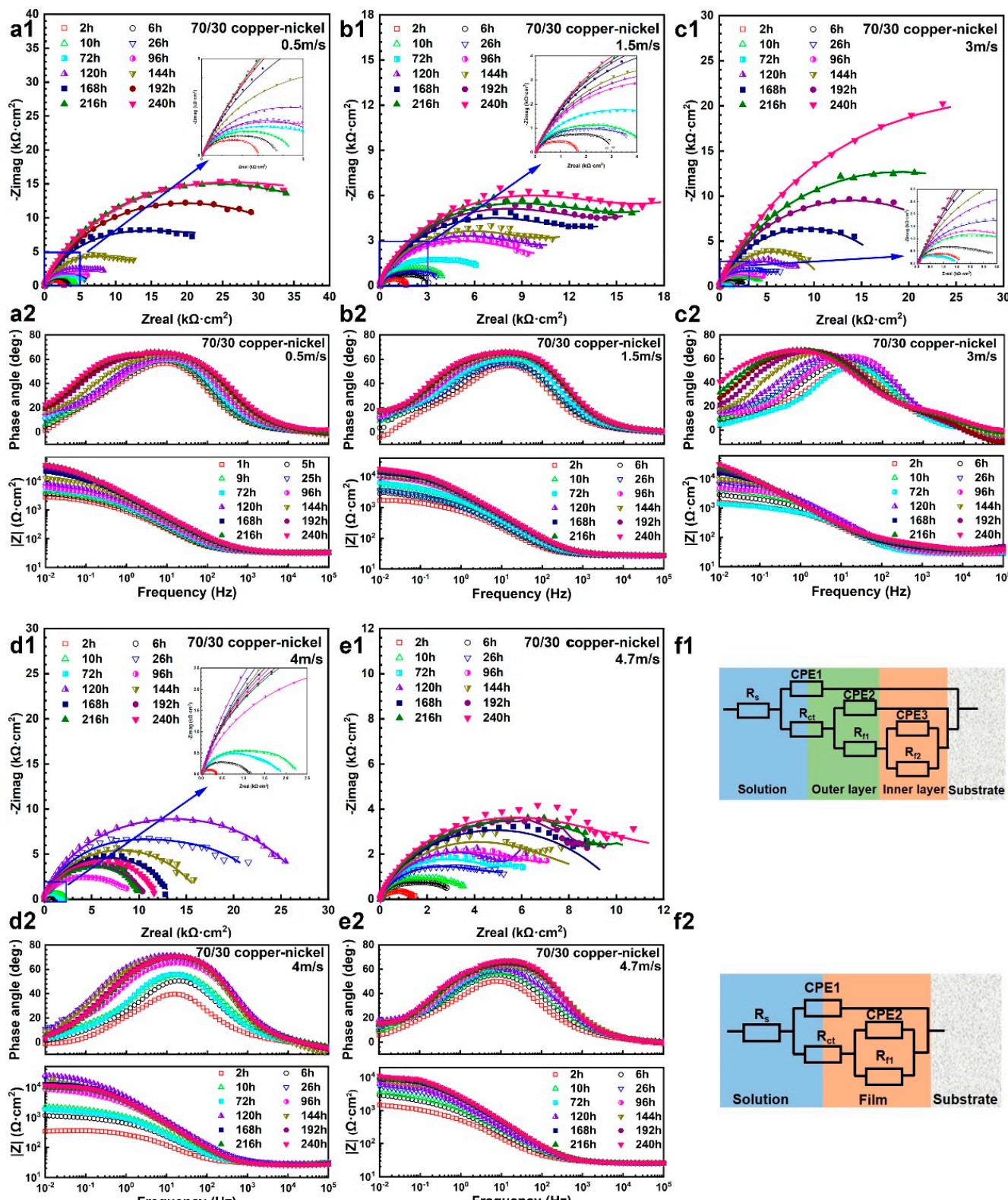

**Figure 3.** Nyquist plots and Bode plots of the 70/30 samples monitored during the 10-day erosion–corrosion tests at the flow velocities of 0.5 m/s (**a1,a2**), 1.5 m/s (**b1,b2**), 3 m/s (**c1,c2**), 4 m/s (**d1,d2**), and 4.7 m/s (**e1,e2**); the equivalent circuits used for EIS fitting tested at lower flow velocities of 0.5 m/s and 1.5 m/s (**f1**) and at higher flow velocities of 3 m/s, 4 m/s, and 4.7 m/s (**f2**).

**Table 2.** Electrochemical parameters of 90/10 copper–nickel samples obtained by fitting EIS curves.

| Velocity (m/s) | Time (h) | $R_s$ ($\Omega \cdot cm^2$) | $R_{ct}$ ($\Omega \cdot cm^2$) | $CPE_1 \times 10^{-4}$ ($F \cdot cm^{-2}$) | $n_1$ | $CPE_2 \times 10^{-4}$ ($F \cdot cm^{-2}$) | $n_2$ | $R_{f1}$ ($\Omega \cdot cm^2$) | $R_{f2}$ ($\Omega \cdot cm^2$) | $CPE_3 \times 10^{-4}$ ($F \cdot cm^{-2}$) | $n_3$ | $\Sigma\chi^2 \times 10^{-3}$ |
|---|---|---|---|---|---|---|---|---|---|---|---|---|
| 0.5 | 1 | 36.87 | 139.8 | 2.82 | 0.75 | 3.43 | 0.57 | 1427 | 1035 | 19.01 | 0.72 | 0.160 |
| | 5 | 36.64 | 147.2 | 3.53 | 0.68 | 0.84 | 0.73 | 2188 | 2312 | 16.44 | 0.69 | 0.511 |
| | 9 | 37.68 | 134.3 | 2.80 | 0.72 | 1.48 | 0.70 | 2112 | 2406 | 14.61 | 0.64 | 0.402 |
| | 25 | 37.60 | 141.8 | 2.56 | 0.72 | 2.34 | 0.71 | 2605 | 2464 | 18.65 | 0.81 | 0.331 |
| | 72 | 35.76 | 142.4 | 2.08 | 0.75 | 3.54 | 0.72 | 2001 | 3490 | 13.62 | 0.76 | 0.774 |
| | 96 | 35.00 | 215.9 | 2.51 | 0.72 | 3.19 | 0.75 | 1944 | 4005 | 14.87 | 0.73 | 0.711 |
| | 120 | 34.76 | 146.8 | 1.66 | 0.78 | 4.21 | 0.71 | 2303 | 5357 | 17.24 | 0.71 | 0.612 |
| | 144 | 34.66 | 142.4 | 1.59 | 0.78 | 4.91 | 0.66 | 2554 | 4182 | 15.74 | 0.65 | 0.579 |
| | 168 | 34.61 | 117.3 | 1.30 | 0.81 | 5.48 | 0.64 | 2823 | 4336 | 13.72 | 0.58 | 0.501 |
| | 192 | 34.26 | 143.5 | 1.55 | 0.78 | 5.40 | 0.65 | 2977 | 4748 | 13.02 | 0.52 | 0.383 |
| | 216 | 33.61 | 145.1 | 1.57 | 0.78 | 5.23 | 0.66 | 2509 | 4123 | 10.89 | 0.56 | 0.468 |
| | 240 | 34.33 | 135.4 | 1.43 | 0.78 | 5.25 | 0.63 | 2528 | 4667 | 12.61 | 0.59 | 0.529 |
| 1.5 | 1 | 28.55 | 107.3 | 3.79 | 0.68 | 0.95 | 0.90 | 1359 | 1291 | 24.61 | 0.58 | 1.553 |
| | 5 | 27.82 | 105.1 | 3.09 | 0.73 | 1.32 | 0.80 | 1148 | 1747 | 6.52 | 0.58 | 0.252 |
| | 9 | 27.78 | 135.6 | 3.78 | 0.71 | 1.99 | 0.75 | 1701 | 1631 | 10.26 | 0.54 | 0.114 |
| | 25 | 28.07 | 111.5 | 3.89 | 0.69 | 3.02 | 0.56 | 1322 | 1464 | 13.93 | 0.58 | 0.193 |
| | 72 | 28.92 | 153.7 | 2.49 | 0.73 | 3.11 | 0.76 | 1488 | 3934 | 13.85 | 0.51 | 0.232 |
| | 96 | 29.43 | 111.9 | 2.02 | 0.74 | 3.75 | 0.69 | 1750 | 4781 | 5.95 | 0.54 | 0.411 |
| | 120 | 31.03 | 116.3 | 1.94 | 0.76 | 6.59 | 0.68 | 1086 | 6510 | 6.00 | 0.74 | 0.197 |
| | 144 | 31.88 | 105.0 | 1.44 | 0.78 | 5.22 | 0.67 | 1481 | 7320 | 4.21 | 0.73 | 0.702 |
| | 168 | 32.62 | 80.2 | 1.12 | 0.81 | 5.60 | 0.67 | 1603 | 7200 | 4.05 | 0.78 | 0.891 |
| | 192 | 33.23 | 138.2 | 2.25 | 0.72 | 4.35 | 0.74 | 1682 | 8164 | 4.87 | 0.76 | 0.667 |
| | 216 | 33.81 | 137.3 | 2.41 | 0.71 | 4.26 | 0.75 | 1894 | 8586 | 4.82 | 0.79 | 0.741 |
| | 240 | 34.77 | 101.1 | 1.83 | 0.73 | 4.85 | 0.71 | 2092 | 8829 | 4.37 | 0.83 | 0.965 |
| 3 | 1 | 28.77 | 15.79 | 6.05 | 0.66 | 3.22 | 0.66 | 1856 | | | | 0.240 |
| | 5 | 28.71 | 10.75 | 0.51 | 0.90 | 6.30 | 0.63 | 2392 | | | | 0.506 |
| | 9 | 28.59 | 12.62 | 0.57 | 0.90 | 6.77 | 0.62 | 2696 | | | | 0.557 |
| | 25 | 28.43 | 84.64 | 3.68 | 0.70 | 2.81 | 0.58 | 3010 | | | | 0.682 |
| | 72 | 28.63 | 42.70 | 1.96 | 0.75 | 3.75 | 0.59 | 4213 | | | | 0.472 |
| | 96 | 29.94 | 53.27 | 2.32 | 0.71 | 3.24 | 0.58 | 5382 | | | | 0.483 |
| | 120 | 30.04 | 43.20 | 1.91 | 0.71 | 3.46 | 0.59 | 5762 | | | | 0.488 |
| | 144 | 29.80 | 52.47 | 2.89 | 0.69 | 2.29 | 0.62 | 4918 | | | | 1.020 |
| | 168 | 29.41 | 31.09 | 2.55 | 0.67 | 3.06 | 0.61 | 6462 | | | | 0.553 |
| | 192 | 29.66 | 45.22 | 3.39 | 0.63 | 2.30 | 0.63 | 7407 | | | | 0.596 |
| | 216 | 29.81 | 49.65 | 3.37 | 0.62 | 2.32 | 0.64 | 8299 | | | | 0.633 |
| | 240 | 26.51 | 52.31 | 5.12 | 0.58 | 1.89 | 0.66 | 7768 | | | | 1.510 |
| 4 | 1 | 31.38 | 27.25 | 4.94 | 0.68 | 3.03 | 0.08 | 680 | | | | 1.200 |
| | 5 | 29.9 | 16.45 | 0.99 | 0.85 | 6.19 | 0.51 | 823 | | | | 0.179 |
| | 9 | 30.04 | 25.28 | 0.90 | 0.88 | 4.81 | 0.51 | 975 | | | | 0.1.94 |
| | 25 | 30.26 | 30.12 | 1.06 | 0.87 | 5.45 | 0.53 | 857 | | | | 0.316 |
| | 72 | 29.31 | 23.40 | 0.56 | 0.90 | 5.65 | 0.56 | 1000 | | | | 0.923 |
| | 96 | 29.04 | 26.36 | 0.44 | 0.66 | 1.18 | 0.58 | 1188 | | | | 0.684 |
| | 120 | 29.08 | 23.84 | 1.85 | 0.77 | 7.73 | 0.51 | 1042 | | | | 1.870 |
| | 144 | 27.04 | 25.50 | 1.73 | 0.75 | 5.70 | 0.52 | 1807 | | | | 0.744 |
| | 168 | 27.93 | 23.03 | 0.66 | 0.87 | 8.22 | 0.57 | 2658 | | | | 0.501 |
| | 192 | 28.04 | 27.38 | 0.82 | 0.84 | 8.89 | 0.57 | 3655 | | | | 0.340 |
| | 216 | 28.48 | 30.20 | 0.93 | 0.82 | 9.12 | 0.57 | 4680 | | | | 0.309 |
| | 240 | 22.42 | 39.25 | 2.49 | 0.71 | 7.81 | 0.58 | 4922 | | | | 1.200 |
| 4.7 | 1 | 26.49 | 156.8 | 9.11 | 0.64 | 0.12 | 1.00 | 512 | | | | 1.14 |
| | 5 | 26.2 | 122.6 | 7.73 | 0.62 | 0.65 | 1.00 | 1231 | | | | 1.16 |
| | 9 | 25.85 | 122.3 | 3.44 | 0.71 | 4.69 | 0.52 | 1449 | | | | 0.906 |
| | 25 | 25.86 | 142.0 | 4.66 | 0.67 | 3.88 | 0.56 | 1771 | | | | 0.736 |
| | 72 | 25.74 | 151.7 | 4.99 | 0.66 | 3.35 | 0.53 | 1573 | | | | 1.705 |
| | 96 | 25.94 | 153.9 | 7.14 | 0.62 | 2.12 | 0.34 | 1937 | | | | 1.18 |
| | 120 | 25.74 | 179.4 | 7.15 | 0.61 | 2.35 | 0.18 | 2172 | | | | 1.14 |
| | 144 | 26.11 | 119.0 | 6.78 | 0.62 | 3.77 | 0.52 | 2007 | | | | 1.19 |
| | 168 | 25.84 | 127.1 | 7.66 | 0.60 | 3.04 | 0.58 | 2482 | | | | 0.968 |
| | 192 | 26.42 | 128.9 | 6.87 | 0.61 | 3.55 | 0.62 | 2849 | | | | 0.910 |
| | 216 | 26.79 | 130.0 | 5.50 | 0.64 | 4.75 | 0.61 | 3104 | | | | 1.10 |
| | 240 | 25.64 | 137.3 | 6.18 | 0.61 | 3.80 | 0.62 | 3440 | | | | 1.07 |

**Table 3.** Electrochemical parameters of 70/30 copper–nickel samples obtained by fitting EIS curves.

| Velocity (m/s) | Time (h) | $R_s$ ($\Omega \cdot cm^2$) | $R_{ct}$ ($\Omega \cdot cm^2$) | $CPE_1 \times 10^{-4}$ ($F \cdot cm^{-2}$) | $n_1$ | $CPE_2 \times 10^{-4}$ ($F \cdot cm^{-2}$) | $n_2$ | $R_{f1}$ ($\Omega \cdot cm^2$) | $R_{f2}$ ($\Omega \cdot cm^2$) | $CPE_3 \times 10^{-4}$ ($F \cdot cm^{-2}$) | $n_3$ | $\Sigma\chi^2 \times 10^{-3}$ |
|---|---|---|---|---|---|---|---|---|---|---|---|---|
| 0.5 | 2 | 34.09 | 149.6 | 0.99 | 0.88 | 1.23 | 0.65 | 1411 | 1239 | 5.93 | 0.73 | 0.405 |
| | 6 | 33.89 | 207.5 | 1.23 | 0.83 | 0.14 | 0.51 | 1167 | 2364 | 3.92 | 0.62 | 0.272 |
| | 10 | 34.06 | 176.3 | 0.92 | 0.87 | 0.67 | 0.51 | 1328 | 3390 | 3.04 | 0.55 | 1.88 |
| | 26 | 34.08 | 178.8 | 0.51 | 0.92 | 0.64 | 0.78 | 1081 | 4972 | 1.54 | 0.53 | 1.151 |
| | 72 | 31.94 | 189 | 0.89 | 0.83 | 0.56 | 0.64 | 3169 | 2072 | 8.66 | 0.78 | 0.761 |
| | 96 | 31.46 | 128.7 | 0.71 | 0.84 | 0.55 | 0.66 | 3661 | 3072 | 6.89 | 0.66 | 0.683 |
| | 120 | 31.42 | 133.4 | 0.74 | 0.82 | 0.22 | 0.72 | 3518 | 5479 | 2.09 | 0.56 | 0.915 |
| | 144 | 32.18 | 177.2 | 0.55 | 0.84 | 0.21 | 0.79 | 3637 | 11,510 | 0.79 | 0.59 | 1.001 |
| | 168 | 28.82 | 110.1 | 0.39 | 0.84 | 0.26 | 0.78 | 6404 | 22,500 | 0.69 | 0.57 | 1.068 |
| | 192 | 32.86 | 162.9 | 0.53 | 0.8 | 0.14 | 0.77 | 7862 | 29,870 | 0.27 | 0.55 | 0.195 |
| | 216 | 32.64 | 125.3 | 0.42 | 0.81 | 0.19 | 0.75 | 8367 | 35,490 | 0.23 | 0.63 | 0.272 |
| | 240 | 33.13 | 165.6 | 0.47 | 0.79 | 0.13 | 0.77 | 8520 | 40,830 | 0.27 | 0.52 | 0.681 |
| 1.5 | 2 | 27.05 | 40.65 | 1.48 | 0.81 | 0.63 | 0.5 | 986 | 624 | 7.33 | 0.88 | 0.268 |
| | 6 | 27.28 | 56.83 | 1.14 | 0.81 | 0.35 | 0.7 | 1622 | 1608 | 11.63 | 0.67 | 0.106 |
| | 10 | 27.14 | 52.17 | 1.39 | 0.78 | 0.32 | 0.66 | 2460 | 2032 | 9.03 | 0.63 | 0.065 |
| | 26 | 27.07 | 50.99 | 0.62 | 0.86 | 0.47 | 0.81 | 744 | 3882 | 3.68 | 0.5 | 0.754 |
| | 72 | 27.96 | 54.22 | 0.53 | 0.83 | 0.24 | 0.8 | 1157 | 5208 | 1.47 | 0.55 | 0.976 |
| | 96 | 28 | 48.26 | 0.49 | 0.83 | 0.15 | 0.81 | 3150 | 8186 | 1.31 | 0.54 | 0.485 |
| | 120 | 31.19 | 52.08 | 0.5 | 0.91 | 7.87 | 0.58 | 2224 | 12,690 | 1.22 | 0.6 | 0.731 |
| | 144 | 28.55 | 56.22 | 0.24 | 0.9 | 0.19 | 0.87 | 1254 | 13,550 | 0.92 | 0.57 | 2.4 |
| | 168 | 28.53 | 57.58 | 0.29 | 0.87 | 0.17 | 0.84 | 1649 | 16,010 | 0.7 | 0.52 | 1.135 |
| | 192 | 28.26 | 51.82 | 0.28 | 0.86 | 0.15 | 0.86 | 1986 | 19,640 | 0.71 | 0.52 | 1.952 |
| | 216 | 27.67 | 47.31 | 0.22 | 0.88 | 0.15 | 0.87 | 1445 | 21,360 | 0.64 | 0.53 | 1.943 |
| | 240 | 27.81 | 52.73 | 0.21 | 0.89 | 0.17 | 0.85 | 2083 | 24,050 | 0.66 | 0.53 | 2.635 |
| 3 | 2 | 27.88 | 82.92 | 1.11 | 0.86 | 3.33 | 0.54 | 1499 | | | | 0.448 |
| | 6 | 27.97 | 73.61 | 0.75 | 0.86 | 1.98 | 0.53 | 3372 | | | | 0.515 |
| | 10 | 28.03 | 82.32 | 0.34 | 0.94 | 2.23 | 0.55 | 6038 | | | | 0.918 |
| | 26 | 28.68 | 66.51 | 0.31 | 0.93 | 1.2 | 0.59 | 6487 | | | | 1.5 |
| | 72 | 28.01 | 76.52 | 0.46 | 0.95 | 2.89 | 0.57 | 1220 | | | | 2.83 |
| | 96 | 28.74 | 65.88 | 0.25 | 0.97 | 1.58 | 0.55 | 6434 | | | | 0.704 |
| | 120 | 31.67 | 74.43 | 0.32 | 0.91 | 1.14 | 0.63 | 10,390 | | | | 1.933 |
| | 144 | 34.3 | 24.75 | 0.14 | 0.97 | 1.49 | 0.76 | 11,310 | | | | 1.08 |
| | 168 | 38.71 | 29.44 | 0.11 | 1 | 1.53 | 0.77 | 18,370 | | | | 4.092 |
| | 192 | 40.67 | 37.28 | 0.12 | 0.97 | 1.54 | 0.77 | 24,970 | | | | 4.951 |
| | 216 | 37.42 | 56.16 | 0.46 | 0.78 | 1.22 | 0.79 | 30,280 | | | | 1.753 |
| | 240 | 37.92 | 66.86 | 0.44 | 0.75 | 1.27 | 0.79 | 40,300 | | | | 2.994 |
| 4 | 2 | 29.03 | 26.92 | 1.28 | 0.87 | 4.17 | 0.62 | 312 | | | | 0.971 |
| | 6 | 28.66 | 36.17 | 0.61 | 0.89 | 2.66 | 0.54 | 1069 | | | | 1.102 |
| | 10 | 28.75 | 34.55 | 0.5 | 0.87 | 1.81 | 0.52 | 2221 | | | | 0.813 |
| | 26 | 29.76 | 25.97 | 0.33 | 0.86 | 0.3 | 0.54 | 23,360 | | | | 1.714 |
| | 72 | 28.6 | 22.89 | 0.4 | 0.94 | 2.56 | 0.53 | 2083 | | | | 0.355 |
| | 96 | 28.59 | 25.24 | 0.33 | 0.88 | 0.63 | 0.53 | 9110 | | | | 0.873 |
| | 120 | 28.18 | 35.18 | 0.24 | 0.9 | 0.3 | 0.54 | 32,460 | | | | 0.573 |
| | 144 | 27.44 | 48.29 | 0.24 | 0.92 | 0.39 | 0.59 | 17,360 | | | | 2.121 |
| | 168 | 26.66 | 34.65 | 0.13 | 0.98 | 0.46 | 0.69 | 13,520 | | | | 1.03 |
| | 192 | 25.97 | 23.9 | 0.25 | 0.92 | 0.39 | 0.62 | 11,050 | | | | 0.721 |
| | 216 | 25.69 | 24.68 | 0.3 | 0.9 | 0.35 | 0.59 | 10,630 | | | | 0.762 |
| | 240 | 22.08 | 34.81 | 0.25 | 0.93 | 0.58 | 0.52 | 13,510 | | | | 1.433 |
| 4.7 | 2 | 26.3 | 30.25 | 3.3 | 0.73 | 3.62 | 0.23 | 1838 | | | | 0.766 |
| | 6 | 26.35 | 32.18 | 1.33 | 0.79 | 2.05 | 0.51 | 3037 | | | | 1.404 |
| | 10 | 25.83 | 32.73 | 1.06 | 0.79 | 1.29 | 0.58 | 3276 | | | | 1.68 |
| | 26 | 25.87 | 31.74 | 0.47 | 0.88 | 1.35 | 0.56 | 5482 | | | | 1.488 |
| | 72 | 25.58 | 31.77 | 0.55 | 0.86 | 0.96 | 0.52 | 6575 | | | | 0.713 |
| | 96 | 25.75 | 31.51 | 0.5 | 0.87 | 0.89 | 0.53 | 7812 | | | | 1.15 |
| | 120 | 25.68 | 36.19 | 1.54 | 0.75 | 1.74 | 0.57 | 9404 | | | | 1.152 |
| | 144 | 25.32 | 37.64 | 0.49 | 0.86 | 1.04 | 0.53 | 10,400 | | | | 0.799 |
| | 168 | 24.98 | 37.6 | 0.52 | 0.84 | 0.6 | 0.54 | 10,510 | | | | 1.312 |
| | 192 | 25.8 | 31.13 | 0.49 | 0.85 | 0.54 | 0.52 | 10,940 | | | | 0.893 |
| | 216 | 25.72 | 32.24 | 0.44 | 0.87 | 0.56 | 0.54 | 12,040 | | | | 0.182 |
| | 240 | 25.42 | 34.28 | 0.46 | 0.86 | 0.48 | 0.54 | 14,320 | | | | 0.572 |

Meanwhile, the equivalent capacitance and equivalent film capacitance of the cuprous oxide layer were calculated with the utilization of Equation (1) [26].

$$C_{eff} = Q^{1/n} R_f^{(1-n)/n} \tag{1}$$

where $C_{eff}$ is the calculated film capacitance, $Q$ represents the impedance of CPE, n is the empirical exponent, which can vary between 1 for a perfect capacitor and 0 for a perfect

resistor, and $R_f$ is the film resistance. Moreover, the film capacitance is related to the equivalent film thickness in Equation (2) [27].

$$C_{eff} = \frac{\varepsilon \varepsilon_0 A}{d} \qquad (2)$$

where $\varepsilon$ is the vacuum permittivity ($8.8542 \times 10^{-12}$ F m$^{-1}$), $\varepsilon_0$ is the dielectric constant (7.11 for cuprous (I) oxide [28]), $A$ is the effective surface area of the sample, and $d$ is the film thickness (m). Then, combining Equations (1) and (2), the equivalent film thickness can be expressed as Equation (3).

$$d = \frac{\varepsilon \varepsilon_0 A}{Q^{1/n} R_f^{(1-n)/n}} \qquad (3)$$

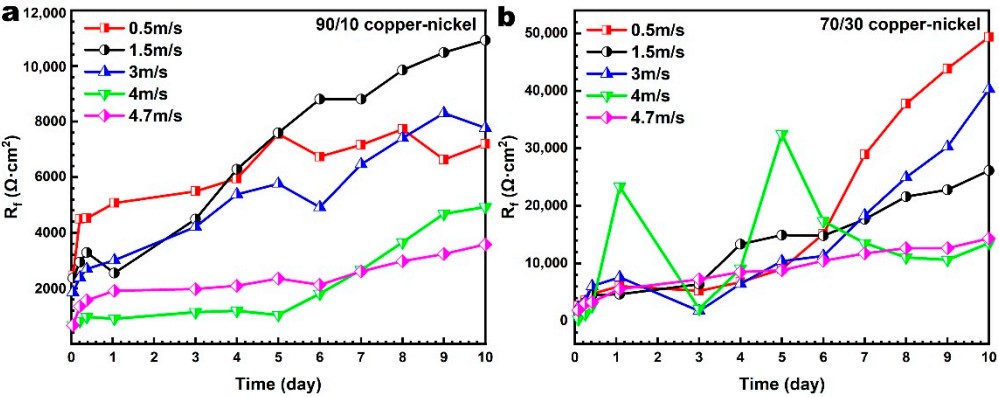

**Figure 4.** Evolution of the film resistance ($R_f = R_{f1} + R_{f2}$) collected from Tables 2 and 3 with erosion–corrosion time at different flow velocities for the 90/10 copper–nickel sample (**a**) and the 70/30 copper–nickel sample (**b**), respectively.

It should be noted that only the thickness of the cuprous oxide film is obtained here, and the thickness of the Cu$_2$(OH)$_3$Cl outer layer is not calculated because the corrosion resistance of copper–nickel alloys is mainly dependent on the cuprous oxide film [4,5,29]. Specifically, CPE$_2$, n$_2$, R$_{f1}$ in Tables 2 and 3 are used for both 90/10 copper–nickel and 70/30 copper–nickel samples tested at 3 m/s, 4 m/s, and 4.7 m/s; CPE$_3$, n$_3$, and R$_{f2}$ in Tables 2 and 3 are used for both 90/10 copper–nickel and 70/30 copper–nickel samples tested at 0.5 m/s and 1.5 m/s. Except for very few data, the C$_{eff}$ of 90/10 copper–nickel sample ranges between 150–7000 µF/cm$^2$ and the corresponding equivalent film thickness (d) ranges between 0.03–1.37 µm; the C$_{eff}$ of 70/30 copper–nickel sample ranges between 17–1583 µF/cm$^2$ and the corresponding equivalent film thickness ranges between 0.12–11.22 µm. Comparing the C$_{eff}$ and d of the 90/10 copper–nickel sample with that of the 70/30 copper–nickel sample, we can see that the 70/30 copper–nickel sample has a lower C$_{eff}$ value and higher d value, which can explain the better corrosion resistance of the 70/30 copper–nickel sample.

### 3.1.2. Potentiodynamic Polarization Curves

Potentiodynamic polarization curves of 90/10 and 70/30 copper–nickel samples tested in 1 wt% NaCl solution at different flow velocities are displayed in Figure 5. As shown in Figure 5a, for the 90/10 copper–nickel samples tested after the 10-day erosion–corrosion test at lower flow velocities of 0.5 m/s and 1.5 m/s, a low-current-density region is recorded in the anodic polarization branches, indicating that passivity is established on the sample surface, which may be attributed to the protective corrosion product film formed during the 10-day erosion–corrosion test [29]. However, it is not a typical passivation region because the passive current density $I_p$ increases with the applied potential shifting positively, while in a typical passivation region of stainless steels and other passive alloys, $I_p$ basically remains constant. When the flow velocity increases from 1.5 m/s to 3 m/s, the corrosion

potential shifts negatively, and both the anodic and cathodic current densities increase, but the anodic process is accelerated more significantly. When the flow velocity further increases to 4 m/s and 4.7 m/s, passivity cannot be maintained and active dissolution dominates the anodic behavior of the samples after the 10-day erosion–corrosion test, which indicates no protective film is formed on the substrate [30]. Meanwhile, the cathodic process is accelerated dramatically and the corrosion potential shifts positively, suggesting that the presence of the corrosion product film mainly inhibits the cathodic process. For the freshly polished 90/10 samples in Figure 5a, typical active dissolution behavior is observed in the anodic branch at each flow velocity. Additionally, as we can see, both anodic and cathodic branches shift slightly in the positive direction with the increased flow velocity. Compared with the polarization curves of the 90/10 samples tested after the 10-day erosion–corrosion test, the corrosion potential of the freshly polished 90/10 samples shifts positively, and the cathodic current density increases significantly.

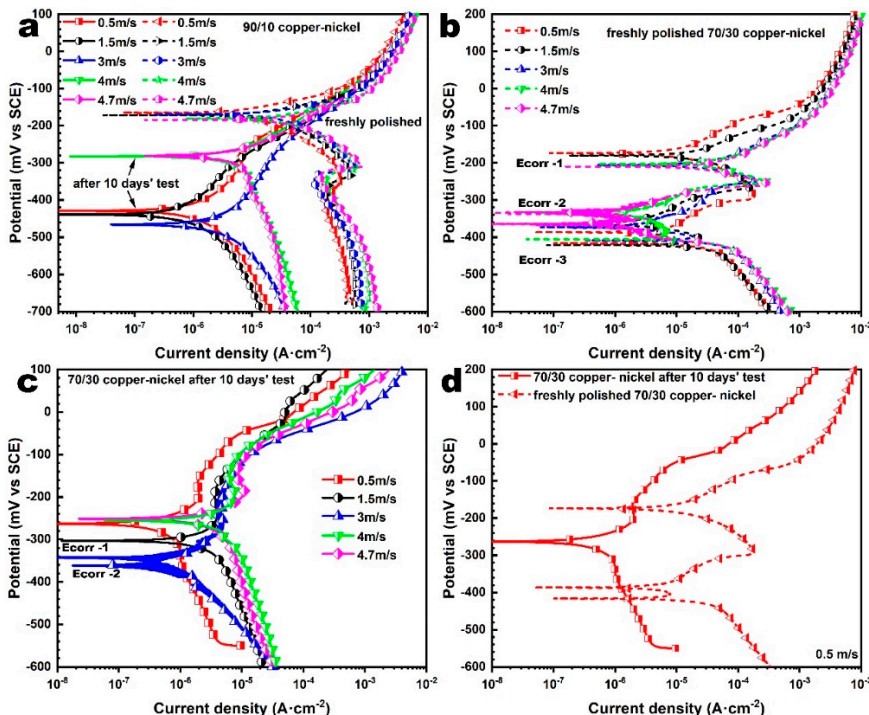

**Figure 5.** Potentiodynamic polarization curves of 90/10 and 70/30 samples tested in 1 wt% NaCl solution at different flow velocities: (**a**) for the freshly polished 90/10 samples as well as the samples after the 10-day erosion–corrosion test at different flow velocities; (**b**) for the freshly polished 70/30 samples; (**c**) for the 70/30 samples after the 10-day erosion–corrosion test at different flow velocities; and (**d**) a comparison of the polarization curve of the 70/30 sample after the 10-day erosion–corrosion test at 0.5 m/s with that of the freshly polished 70/30 sample.

For the 70/30 copper–nickel samples after the 10-day erosion–corrosion test at different flow velocities, as shown in Figure 5c, low current density plateaus ($10^{-6}$~$10^{-5}$ μA·cm$^{-2}$) appear in the anodic polarization branches, which demonstrates a passive film is present on the substrate and spontaneous passivation dominates the anodic behavior. When the flow velocity increases from 0.5 m/s to 1.5 m/s and 3 m/s, the corrosion potential shifts negatively, and both anodic and cathodic current densities increase, indicating that the electrode becomes more active compared to that at 0.5 m/s. With further increase of the flow velocity to 4 m/s and 4.7 m/s, the corrosion potential shifts positively, both anodic and cathodic current densities increase, and the passivation region is narrowed compared with that at lower flow velocities. Additionally, 2 corrosion potentials appear in the polarization curve at 3 m/s, which demonstrates that the 70/30 copper–nickel sample is in an unstable state at 3 m/s [31]. For the freshly polished 70/30 copper–nickel samples in

Figure 5b, 3 corrosion potentials are recorded in the polarization curves tested at all the flow velocities. The phenomenon of multiple corrosion potentials is reported in chromium [32], high-nitrogen bearing stainless steel [33] in acidic chloride solution and pure titanium in fluoride-containing sulfuric acid [34], and is explained using the ideal polarization curve models in an unstable system. Figure 5d displays the comparison of the polarization curve of the 70/30 sample after the 10-day erosion–corrosion test at 0.5 m/s with that of the freshly polished 70/30 sample tested at the same flow velocity. Apparently, both anodic and cathodic reactions are suppressed after the 10-day erosion–corrosion test.

As we can see in Figure 5, the polarization curves of the 90/10 and 70/30 samples tested in different states have different characteristics, so different fitting methods are used to obtain the corrosion current density. For the polarization curves of the freshly polished 90/10 samples, anodic Tafel fitting is used because the anodic Tafel region is present at a potential higher than +100 mV vs. OCP while the cathodic Tafel region is absent. For the samples after the 10-day erosion–corrosion test, cathodic Tafel fitting is used because a single straight Tafel region exists at the cathodic potential lower than $-100$ mV vs. OCP, while a single straight Tafel region is absent at the anodic potential higher than +100 mV vs. OCP. For the freshly polished 70/30 samples, low-polarization fitting is conducted in the potential range of $-10 \sim -30$ mV vs. $E_{corr-1}$ because neither the anodic nor the cathodic Tafel region can be found in the polarization curve with three corrosion potentials. For the 70/30 samples after the 10-day erosion–corrosion test, corrosion current density is obtained from the low current density plateau in the anodic branch as the samples show spontaneous passivation behavior. The corrosion potentials and current densities obtained from Figure 5 are shown in Table 4.

**Table 4.** Electrochemical parameters obtained from the fitting of the polarization curves of 90/10 copper–nickel alloys and 70/30 copper–nickel alloys.

| Sample Status | Flow Velocity (m/s) | $E_{corr}$ (mV vs. SCE) | $I_{corr}$ ($\mu A \cdot cm^{-2}$) |
|---|---|---|---|
| Freshly polished 90/10 copper–nickel samples | 0.5 | −160.5 | 18.16 |
| | 1.5 | −171.5 | 43.53 |
| | 3 | −169.3 | 61.47 |
| | 4 | −180.8 | 84.13 |
| | 4.7 | −184.3 | 90.93 |
| 90/10 copper–nickel samples after the 10-day erosion–corrosion test | 0.5 | −429.0 | 3.85 |
| | 1.5 | −438.9 | 2.39 |
| | 3 | −465.8 | 8.54 |
| | 4 | −282.5 | 11.16 |
| | 4.7 | −280.3 | 13.44 |
| Freshly polished 70/30 copper–nickel samples | 0.5 | −173.0 −386.4 −416.4 | 16.57 |
| | 1.5 | −180.3 −364.4 −421.4 | 28.99 |
| | 3 | −206.2 −373.0 −408.7 | 47.02 |
| | 4 | −203.0 −332.6 −405.6 | 51.37 |
| | 4.7 | −209.9 −334.8 −364.4 | 52.86 |
| 70/30 copper–nickel samples after the 10-day erosion–corrosion test | 0.5 | 263.0 | 1.60 |
| | 1.5 | −303.2 | 3.56 |
| | 3 | −342.3 −361.7 | 4.33 |
| | 4 | −255.8 | 6.62 |
| | 4.7 | −251.5 | 8.49 |

As can be seen in Table 4, for the freshly polished 90/10 copper–nickel samples, the corrosion current density ($I_{corr}$) increases with the increasing flow velocity, but the

corrosion potential ($E_{corr}$) shows an opposite trend. For the 90/10 copper–nickel samples after the 10-day erosion–corrosion test, the $I_{corr}$ follows the sequence of $I_{corr\ 1.5\ m/s} < I_{corr\ 0.5\ m/s} < I_{corr\ 3\ m/s} < I_{corr\ 4.7\ m/s} \approx I_{corr\ 4\ m/s}$, which is consistent with the EIS results suggesting that the film formed at 1.5 m/s shows the best corrosion resistance. The $E_{corr}$ follows the sequence of $E_{corr\ 3\ m/s} < E_{corr\ 1.5\ m/s} < E_{corr\ 0.5\ m/s} < E_{corr\ 4.7\ m/s} \approx E_{corr\ 4\ m/s}$. The samples tested at 4 m/s and 4.7 m/s show a more positive $E_{corr}$ compared to those tested at lower flow velocities. Moreover, the 90/10 copper–nickel samples after the 10-day erosion–corrosion test show more negative $E_{corr}$ and lower $I_{corr}$ than the freshly polished 90/10 copper–nickel samples, which indicates that the protective corrosion product film formed on the 90/10 copper–nickel samples during the erosion–corrosion test mainly inhibits the cathodic process.

For the freshly polished 70/30 copper–nickel samples, the corrosion current density ($I_{corr}$) also increases with the increasing flow velocity. For the 70/30 copper–nickel samples after the 10-day erosion–corrosion test, the fitted $I_{corr}$ in Table 4 follows the sequence of $I_{corr\ 0.5\ m/s} < I_{corr\ 3\ m/s} < I_{corr\ 1.5\ m/s} < I_{corr\ 4\ m/s} < I_{corr\ 4.7\ m/s}$, which suggests that the corrosion product film formed at 0.5 m/s is the most resistant to corrosion. Additionally, the $E_{corr}$ follows the sequence of $E_{corr\text{-}1\ 3\ m/s} < E_{corr\text{-}\ 1.5\ m/s} < E_{corr\text{-}\ 0.5\ m/s} < E_{corr\text{-}\ 4\ m/s} \approx E_{corr\text{-}\ 4.7\ m/s}$. Moreover, $I_{corr}$ of the 70/30 samples after the 10-day erosion–corrosion test is lower than that of the freshly polished 70/30 copper–nickel samples, which is similar to the results of the 90/10 copper–nickel samples.

*3.2. Ex Situ Characterization of Corrosion Product Film after Erosion–Corrosion Test*

3.2.1. SEM and EDS Results

Figure 6 shows the surface and cross-section morphologies of the corrosion product films formed on the 90/10 copper–nickel samples after the 10-day erosion–corrosion test at different flow velocities, including the line scan EDS analysis along the red lines in the cross-section images. When the flow velocity is no more than 3 m/s, continuous corrosion product films with a few cracks are formed on the substrate, as shown in Figure 6a–c. The corrosion product film formed at 1.5 m/s is the most compact compared to those formed at 0.5 m/s or 3 m/s, which may be responsible for the highest $R_f$ and the lowest $I_{corr}$ revealed by the electrochemical results in Figure 4a and Table 4. At 4 m/s, as shown in Figure 6d, part of the substrate is covered with corrosion product film and part of the substrate is exposed. At 4.7 m/s, as shown in Figure 6e, no corrosion product film is found and the exposed substrate is corroded severely.

Figure 7 shows the surface and cross-section morphologies of the corrosion product films formed on the 70/30 copper–nickel samples after the 10-day erosion–corrosion at different flow velocities, including the line scan EDS analysis along the red lines in the cross-section images. At 0.5 m/s, as shown in Figure 7a, the corrosion product film is compact and free from cracks, which may be responsible for the highest $R_f$ and the lowest $I_{corr}$ displayed by the electrochemical results in Figure 4b and Table 4. At 1.5 m/s, as shown in Figure 7b, the corrosion product film is compact but contains a few tiny cracks. At 3 m/s, as shown in Figure 7c, the corrosion product seems more compact than that formed at 1.5 m/s, despite the thinned thickness. At 4 m/s and 4.7 m/s, as shown in Figure 7d,e, the corrosion product film is rimous with huge cracks penetrating it, which may be attributed to the internal stress generated during the fast film formation process [35]. However, the corrosion current density is still maintained at a lower value (~8.5 $\mu A\cdot cm^{-2}$) compared to that of the freshly polished samples, as shown in Table 4, indicating the protectiveness of the corrosion product film. Moreover, the thickness of the corrosion product film decreases while the flow velocity increases.

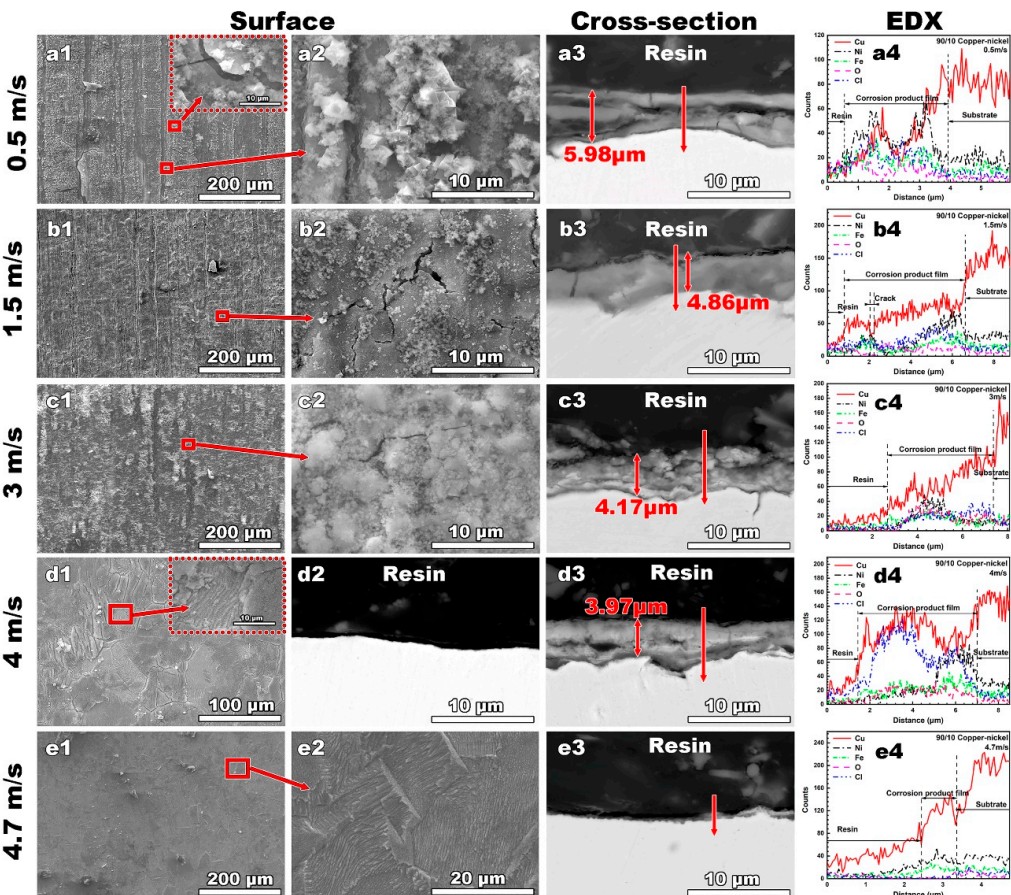

**Figure 6.** Surface and cross-section morphologies of the 90/10 copper–nickel samples after the 10-day erosion–corrosion test in 1 wt% NaCl solution at different flow velocities: (**a1**–**a3**) 0.5 m/s, (**b1**–**b3**) 1.5 m/s, (**c1**–**c3**) 3 m/s, (**d1**–**d3**) 4 m/s, and (**e1**–**e3**) 4.7 m/s; (**a4**–**e4**) the corresponding EDS profiles along the red lines marked in (**a3**–**e3**), respectively.

### 3.2.2. XPS Results

Figure 8 shows the XPS fine spectra of corrosion product films formed on the 90/10 and 70/30 copper–nickel samples after the 10-day erosion–corrosion test at 1.5 m/s. The corresponding Cu 2p spectra of the corrosion product film are shown in Figure 8a. Cu or $Cu_2O$ (the strong Cu 2p3/2 peak at ~932.5 eV) is detected in all the samples. $Cu_2(OH)_3Cl$ or $Cu(OH)_2$ (Cu 2p3/2 peak at 935 eV with shake-up features at ~942.5 eV) are detected only in the middle of but not on the surface of the corrosion product films on both 90/10 and 70/30 samples. Additionally, the copper content in the middle of the film is higher than that on the surface of the film for the 70/30 copper–nickel sample. The corresponding Fe 2p2/3 spectra are shown in Figure 8b. The peak at BE of ~711.5 eV is attributed to γ-FeOOH. However, the Fe content on the surface of the film is higher than that in the middle of the film, which is true for both the 90/10 and the 70/30 copper–nickel samples. The Ni 2p3/2 spectra of different samples are shown in Figure 8c. The peak at BE of ~855.5 eV with a shake-up peak at BE of 861.5 eV is assigned to NiO, the peak at BE of ~856.6 eV with a shake-up peak at BE of ~862.2 eV is assigned to $Ni(OH)_2$, and the peak at BE of ~852.8 eV in the 70/30 sample is assigned to metallic nickel. Additionally, it can be seen that the peak intensity in the middle of the corrosion product film is higher than that on the surface of the corrosion product film for both the 90/10 and 70/30 copper–nickel samples, indicating that the inner layer of the corrosion product film contains more metallic Ni or nickel compound.

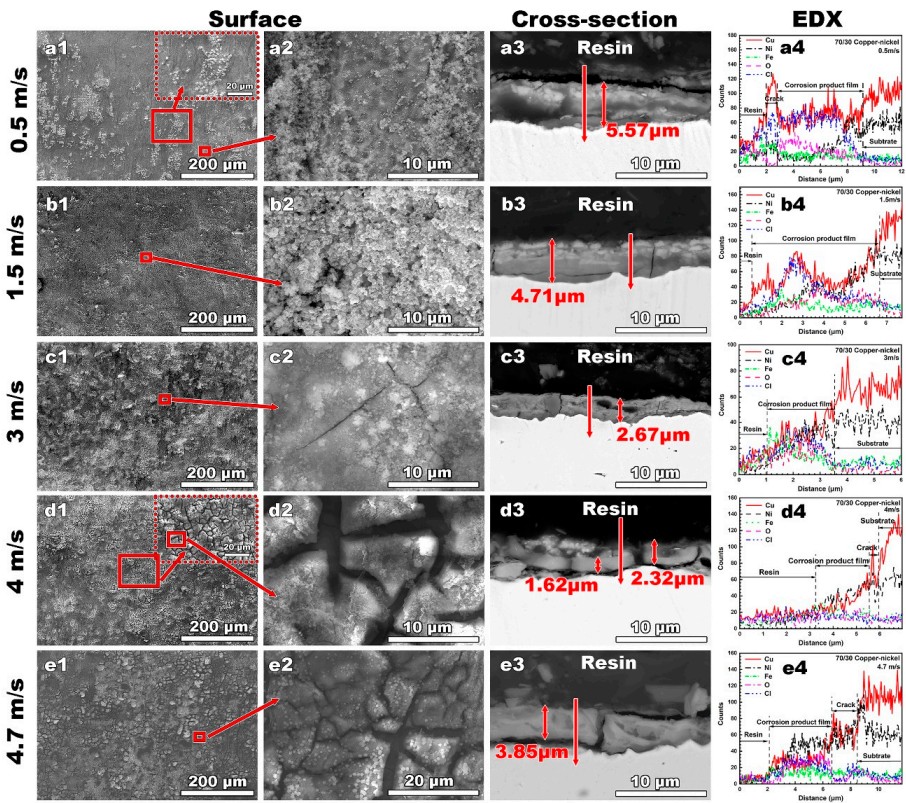

**Figure 7.** Surface and cross-section morphologies of the 70/30 copper–nickel sample after the 10-day erosion–corrosion test in 1% NaCl solution at different flow velocities: (**a1**–**a3**) 0.5 m/s; (**b1**–**b3**) 1.5 m/s; (**c1**–**c3**) 3 m/s; (**d1**–**d3**) 4 m/s; (**e1**–**e3**) 4.7 m/s; (**a4**–**e4**) the corresponding EDS profiles along the red lines marked in (**a3**–**e3**), respectively.

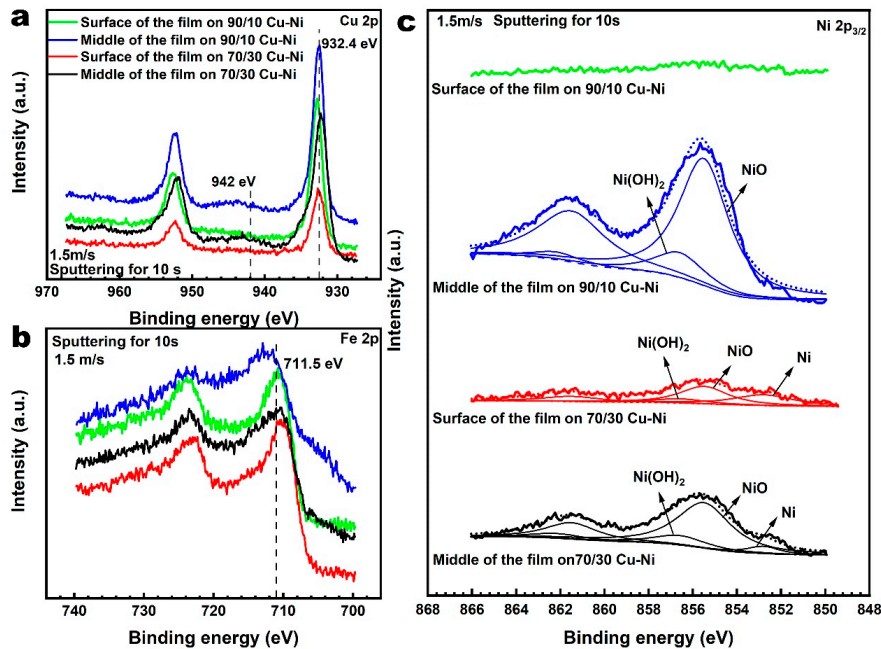

**Figure 8.** Cu 2p spectra (**a**), Fe 2p spectra (**b**), and Ni2p$_{2/3}$ spectra (**c**) after sputtering for 10 s for the 90/10 copper–nickel sample and the 70/30 copper–nickel sample after the 10-day erosion–corrosion test at 1.5 m/s.

## 4. Discussion

### 4.1. Erosion–Corrosion Behavior of the 90/10 Copper–Nickel Tube in Sodium Chloride-Based Fluid

4.1.1. Effect of Corrosion Product Film on Corrosion Behavior

There is an atypical passivation region in the anodic polarization curve of the 90/10 copper–nickel, as shown in Figure 5a, where the passive current density $I_p$ increases with the applied potential shifting positively, while in a typical passivation region of stainless steels and other passive alloys, $I_p$ basically remains constant. This is because the corrosion product film of the 90/10 copper–nickel alloy, which is thick (~µm) and multilayered and takes a long time (several days to 2–3 months) to form and its protectiveness to the substrate, is not so good [2,5], is not a typical passive film which is thin (~nm) and compact and takes a short time (~s) to form and exerts good protectiveness on the substrate [34]. By comparison of the polarization curves of the freshly polished samples with those of the samples after the 10-day erosion–corrosion test in Figure 5a, we can see that the presence of corrosion product film mainly inhibits the cathodic process, i.e., the oxygen reduction reaction (ORR) on the sample surface. The cathodic polarization curves will intersect the anodic polarization curves in different regions with the presence or absence of the corrosion product film on the substrate. From this, an ideal schematic mechanism of the corrosion process is proposed with the potential range of −400 mV–0 mV and the current density range of $10^{-7}$–$10^{-2}$ A cm$^2$ to explain the erosion–corrosion behavior of the 90/10 copper–nickel alloy in 1 wt% NaCl solution at different flow velocities, as shown in Figure 9.

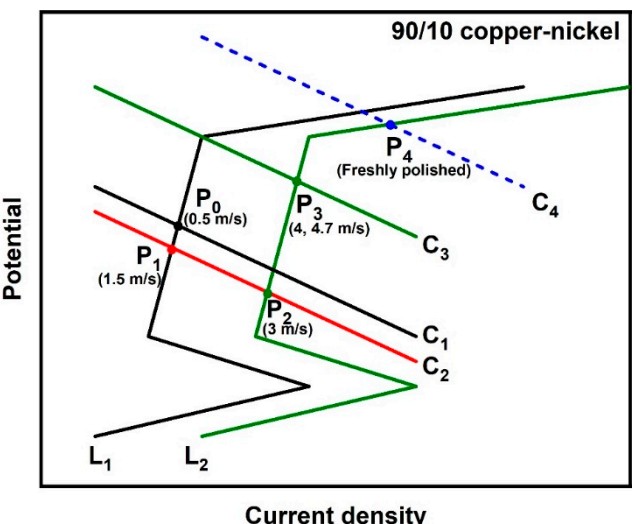

**Figure 9.** An ideal schematic mechanism of the corrosion process of the 90/10 copper–nickel tube in NaCl solution at different flow velocities.

With the presence of a corrosion product film on the substrate, the cathodic polarization curves intersect the anodic polarization curve in the passive region. Curve $L_1$ represents the ideal anodic polarization curves of the 90/10 copper–nickel alloy with a protective corrosion product film formed at low flow velocities (0.5–1.5 m/s). The anodic behavior at 0.5 m/s is similar to that at 1.5 m/s. However, the corrosion product film formed at 1.5 m/s is more compact and more protective than that formed at 0.5 m/s, as supported by the EIS results in Table 2 and the SEM images in Figure 6, which result in a more significant inhibition of the cathodic process compared to that at 0.5 m/s. Therefore, the intersection $P_0$ of $L_1$ and $C_1$ represents the corrosion behavior at 0.5 m/s, and the intersection $P_1$ of $L_1$ and $C_2$ represents the corrosion behavior at 1.5 m/s. At 3 m/s, the protectiveness of the corrosion product film is still present but has obviously deteriorated compared to that at 1.5 m/s. Although both anodic and cathodic reactions are accelerated, the anodic reaction is accelerated more significantly. Therefore, Curve $L_2$ represents the anodic polarization

behavior of the 90/10 copper–nickel alloy with degraded corrosion product film formed at 3 m/s. The intersection $P_2$ of $L_2$ and $C_2$ represents the corrosion behavior at 3 m/s. When the corrosion product film is severely damaged at higher flow velocities of 4 m/s and 4.7 m/s, the cathodic process is dramatically promoted, and the cathodic polarization curves intersect the anodic polarization curves at more positive potentials in the passive region. This results in the more positive $E_{corr}$ and the larger $I_{corr}$. The intersection $P_3$ of $L_2$ and $C_3$ represents the corrosion behavior at 4 m/s and 4.7 m/s. When the corrosion product film is totally absent, as in the case of the freshly polished samples, the cathodic polarization curves intersect the anodic polarization curve in the trans-passive region, which results in the more positive $E_{corr}$ and the larger $I_{corr}$. The intersection $P_4$ of $L_2$ and $C_4$ represents the corrosion behavior of the 90/10 copper–nickel alloy without a corrosion product film on the surface. Additionally, with the flow velocity increasing, the simultaneous acceleration of anodic and cathodic reactions led to the almost unchanged $E_{corr}$ and the increased $I_{corr}$, as can be seen in Figure 5a and Table 4.

#### 4.1.2. Effect of Flow Velocity on Corrosion Product Film

In erosion–corrosion, mass transfer and surface shear stress generated by flowing fluid may have a profound effect on the rate of material degradation, either by modifying the rate of mass transport of chemical species to or from the surface or by shear-stripping films from the metal/solution interface. However, the erosion–corrosion mechanism of copper–nickel alloys is much more complex because the two effects mentioned above are intertwined, which is related to the formation process and characteristics of their corrosion product films in a flowing environment. The effect of flow velocity on the corrosion product film of the 90/10 copper–nickel alloy can be described in Figure 10.

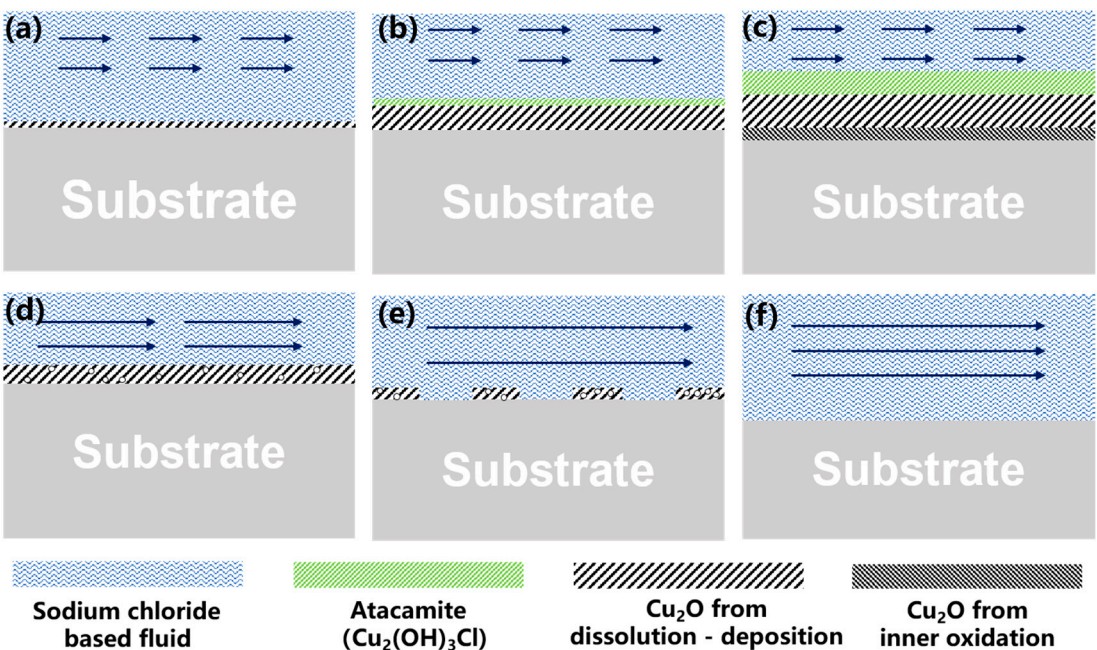

**Figure 10.** Effect of flow velocity on the corrosion product film of the 90/10 copper–nickel alloy in sodium chloride-based fluid: (**a**–**c**) the schematics of the corrosion product film formation process at low flow velocities of 0.5–1.5 m/s; (**d**–**f**) the schematics of the corrosion product films formed at 3 m/s, 4 m/s, and 4.7 m/s, respectively.

When the flow velocity is as low as 0.5–1.5 m/s, as shown in Figure 10a–c, the flowing fluid can promote the formation of a corrosion product film by the enhanced mass transfer. The shear stress generated by flowing fluid at such low velocities is not enough to peel off the corrosion product, although it increases with the increasing flow velocity. In the initial period of exposure, the $Cu_2O$ film is formed on the 90/10 copper–nickel alloy through the

dissolution–repreciptation mechanism described by Equations (4)–(9) [29,36], as shown in Figure 10a. The accompanying cathodic reaction is the oxygen reduction reaction (ORR) through Equation (10).

$$Cu + 2Cl^- \rightarrow CuCl_2^- + e^- \tag{4}$$

$$Cu \rightarrow Cu^+ + e^- \tag{5}$$

$$Cu^+ + 2Cl^- \rightarrow CuCl_2^- \tag{6}$$

$$Cu + Cl^- \rightarrow CuCl + e^- \tag{7}$$

$$CuCl + Cl^- \rightarrow CuCl_2^- \tag{8}$$

$$2CuCl_2^- + 2OH^- \rightarrow Cu_2O + H_2O + 4Cl^- \tag{9}$$

$$O_2 + 2H_2O + 4e^- \rightarrow 4OH^- \tag{10}$$

Since $Cu_2O$ is thermodynamically unstable, with time, prolonged redox transformation of the outmost $Cu_2O$ to atacamite ($Cu_2(OH)_3Cl$) occurs through Equation (11), as displayed in Figure 10b, and atacamite deposits on the surface because of the small shear stress generated by the low flow velocity.

$$Cu_2O + Cl^- + 2H_2O \rightarrow Cu_2(OH)_3Cl + H^+ + 2e^- \tag{11}$$

As mentioned above, the inhibition effect of the corrosion product film on the cathodic process is much greater than that on the anodic process. The corrosion rate increases with increasing the flow velocity in the range studied, as shown in Figures 4a and 5 and Table 4, which reflects that the corrosion rate is controlled at least partially by the rate of mass transfer, i.e., the transport rate of dissolved oxygen diffusing through both the diffusion boundary layer and the corrosion product film, with the corrosion rate being limited by the latter [37]. As the atacamite layer and the porous $Cu_2O$ layer thicken with time, as shown in Figure 10c, the diffusion of dissolved oxygen through the corrosion product film becomes more difficult, so that the growth mode of the $Cu_2O$ film changes from the outward repreciptation to the inward oxidation with a slower rate, as described in our previous research [5,22]. During the inward growth of $Cu_2O$, copper vacancies are generated at the film/solution interface by Equation (12). They then diffuse inward the film, and metallic Cu atoms at the alloy/film interface enter the corrosion product by Equations (13) and (14).

$$Cu_{Cu}^\times(ox) \leftrightarrow Cu^+(aq) + V'_{Cu}(ox) \tag{12}$$

$$Cu(m) + V'_{Cu}(ox) \leftrightarrow Cu_{Cu}^\times(ox) + e' \tag{13}$$

$$e' + h^\bullet(ox) \leftrightarrow 0 \tag{14}$$

where $Cu^\times{}_{Cu}(ox)$ is a copper cation in a regular site of the oxide film, $Cu^+(aq)$ is a dissolved $Cu^+$ ion in seawater, $V'_{Cu}(ox)$ is a negatively charged cation vacancy in the $Cu_2O$ film, and $Cu(m)$ is a copper atom in a regular metal site in the alloy substrate. Consequently, a mature multi-layered corrosion product film formed at low flow velocities on the 90/10 copper–nickel alloy is mainly composed of the atacamite layer, the porous $Cu_2O$ layer from dissolution–repreciptation, and the compact $Cu_2O$ layer from inward oxidation, as shown in Figure 10c.

When the flow velocity increases to 3 m/s, as shown in Figure 10d, the surface shear stress generated by the flow fluid is so high that atacamite cannot deposit on the $Cu_2O$ layer but is carried away by the fluid. So, the $Cu_2O$ layer is exposed to the corrosive fluid, and the dynamic equilibrium between the formation and dissolution of the $Cu_2O$ layer is broken. As a result, the thickness of the $Cu_2O$ layer is reduced with increasing flow velocity, as demonstrated in Figure 6a–c. When the flow velocity increases to 4 m/s, as shown in Figure 10e, the surface shear stress generated by flowing fluid can peel off not only the atacamite layer but also the $Cu_2O$ layer in weak areas, leading to the formation of the partly naked surface in Figure 6d. Moreover, the naked metal area is thermodynamically more

active than the filmed area, which may act as an anode and corrode preferentially. Such an area may be the pitting core and contribute to the pitting perforation that happened in the industrial environment [9]. When the flow velocity increases to 4.7 m/s, as shown in Figure 10f, the surface shear stress generated by flow fluid is high enough to peel the entire corrosion product film off completely, leaving the naked substrate being corroded fast [38], as demonstrated in Figure 6e.

### 4.2. Erosion–Corrosion Behavior of the 70/30 Copper–Nickel Tube in Sodium Chloride-Based Fluid

#### 4.2.1. Effect of Corrosion Product Film on Corrosion Behavior

In the absence of any external polarization, the reduction of depolarizers in corrosive media promotes metal passivation, which is called metal self-passivation or spontaneous passivation. Contrary to the atypical passivation region in the polarization curve of the 90/10 copper–nickel alloy, the 70/30 copper–nickel alloy samples after the 10-day erosion–corrosion test show spontaneous passivation behavior, and typical passivation regions are recorded in their anodic curves, as shown in Figure 5c, indicating that the corrosion product film on the 70/30 copper–nickel alloy is more similar to the passive film compared to that on the 90/10 copper–nickel alloy. By comparison of the polarization curves of the freshly polished 70/30 samples with those of the samples after the 10-day erosion–corrosion test, we can see that the presence of corrosion product film inhibits both anodic and cathodic processes of the 70/30 copper–nickel alloy. However, at different flow velocities and with different surface films formed, the cathodic curve intersects the anodic curve in different regions. Based on this, an ideal schematic mechanism of the corrosion process is proposed with the potential range of $-400$ mV–0 mV and the current density range of $10^{-8}$–$10^{-3}$ A cm$^2$ to explain the erosion–corrosion behavior of the 70/30 copper–nickel alloy in NaCl solution at different flow velocities, as shown in Figure 11.

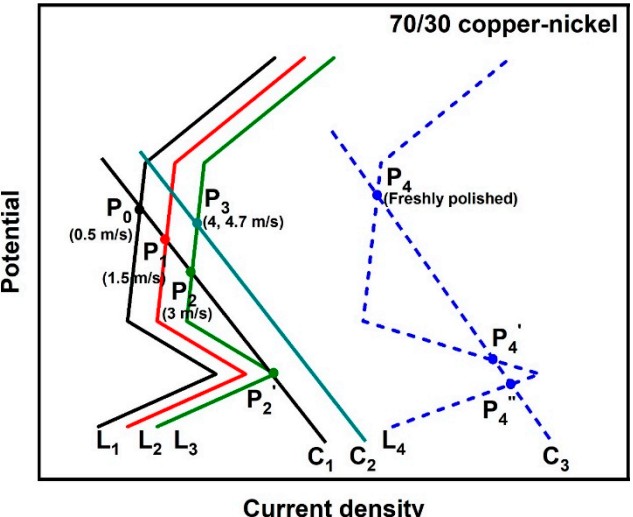

**Figure 11.** An ideal schematic mechanism of the corrosion process of the 70/30 copper–nickel tube in NaCl solution at different flow velocities.

In the low flow velocity range of 0.5–3 m/s with the protective corrosion product film formed, the anodic process of the 70/30 copper–nickel alloy is accelerated more significantly, although both the anodic and cathodic processes are accelerated with the flow velocity increasing. Therefore, in Figure 11, curves $L_1$, $L_2$, and $L_3$ represent the ideal anodic polarization curves of the 70/30 copper–nickel samples after the 10-day erosion–corrosion test at 0.5 m/s, 1.5 m/s, and 3 m/s, respectively, and correspondingly Line $C_1$ represents the ideal cathodic polarization curve in the flow velocity range of 0.5–3 m/s, and their intersections represent the corrosion behavior at 0.5 m/s, 1.5 m/s, and 3 m/s, respectively. Especially, two intersections of $L_3$ and $C_1$ appear in the passive region ($P_2$) and active-

passive region (P$_{2'}$) of the anodic polarization curve, respectively, which stand for the two corrosion potentials of the 70/30 copper–nickel samples after the 10-day erosion–corrosion test at 3 m/s, as shown in Figure 5c. When the flow velocity increases to 4 m/s and 4.7 m/s, the protectiveness of the surface film deteriorates severely, as demonstrated by Figures 4b and 5c and Table 4, and the cathodic process is accelerated more significantly, although both the anodic and cathodic processes are accelerated simultaneously. Therefore, Line C$_2$ represents the ideal cathodic polarization curve of the 70/30 copper–nickel alloy with degraded surface films formed at higher velocities of 4–4.7 m/s, and the intersection P$_3$ of L$_3$ and C$_2$ represents the corrosion behavior of the 70/30 copper–nickel alloy after the 10-day erosion–corrosion test at 4 m/s and 4.7 m/s. At such high flow velocities, the passivation region, although still present, is significantly narrower than that at lower flow velocities, as demonstrated by Figure 5c. When the corrosion product film is totally absent, as in the case of the freshly polished 70/30 samples in Figure 5b, both anodic and cathodic processes are accelerated compared to the cases with the presence of the corrosion product film. Therefore, L$_4$ and C$_3$ represent the ideal anodic and cathodic polarization curves under this condition, respectively. Three intersections of L$_4$ and C$_3$ appear in the active region, active-passive region, and passive region of the anodic curve, respectively, corresponding to the three corrosion potentials of the freshly polished 70/30 samples tested at different flow velocities, as shown in Figure 5b,d.

4.2.2. Effect of Flow Velocity on Corrosion Product Film

Figure 12 shows the effect of flow velocity on the corrosion product film of the 70/30 copper–nickel alloy in a sodium chloride-based fluid schematically. The formation process of corrosion product film on the 70/30 copper–nickel alloy is similar to that on the 90/10 copper–nickel alloy. However, there are also differences, mainly in two aspects. Firstly, from the initial period of exposure, a high concentration of Ni at the substrate/film interface would enter the $Cu_2O$ lattice and occupy the cation vacancies, as shown by Equation (15), which reduces the concentration of Cu vacancies, that is, the ionic conductivity of the $Cu_2O$ film. At the same time, the Ni atom releases two electrons, which in turn neutralize and annihilate the holes in the film, reducing the electronic conductivity of the film, as expressed by Equation (16).

$$Ni(m) + V'_{Cu}(ox) \leftrightarrow Ni^{\bullet}_{Cu}(ox) + 2e' \tag{15}$$

$$2e' + 2h^{\bullet}(ox) \leftrightarrow 0 \tag{16}$$

where *Ni(m)* is a nickel atom in a regular metal site, and *Ni$^{\bullet}$$_{Cu}$(ox)* is a positively charged nickel cation in the cation site of the $Cu_2O$ film. So incorporation of Ni into the bottom $Cu_2O$ film would decrease both the ionic and electronic conductivity of the $Cu_2O$ film and form a compact barrier layer, as shown in Figure 12, which is the reason for the spontaneous passivation behavior and better corrosion resistance of the 70/30 copper–nickel alloy compared to the 90/10 copper–nickel alloy. Secondly, when the flow velocity reaches 4 m/s and even 4.7 m/s, there is still a continuous corrosion product film on the surface of the 70/30 alloy, although the film layer is obviously cracked, as shown in Figure 7d,e. However, the compact barrier layer close to the substrate still exists, as the polarization curve still shows spontaneous passivation behavior, and the corrosion current density ($I_{corr}$) is still at a low value, as demonstrated in Figure 5c and Table 4, which indicates that the surface film still provides good protection to the substrate. The cracks in the film may be attributed to the higher shear stress generated by the flow fluid peeling the outer layer.

*4.3. Comparison of Erosion–Corrosion Behavior for the 90/10 and 70/30 Copper–Nickel Tubes*

As proved by A. M. Beccaria et al. [39], the main difference between the polarization curves of Cu-Ni alloy and pure copper is that a passivation zone appears in the anode polarization part, and the passivation current density decreases with the increase of Ni content. When Ni content reaches 30%, the passivation current density of the alloy is

close to that of pure nickel, and the passivation interval is wide. This is the basis for the design of the 70/30 copper–nickel alloy. As for the main effect of nickel on improving the corrosion resistance of copper alloys, North and Pryor [40] believe that it is mainly the result of nickel being doped into the copper surface film as a dopant. Cuprous oxide is a P-type semiconductor with high defects, and an alloying element (the Ni element is very effective) can be doped into the defective $Cu_2O$ lattice by occupying the cationic vacancy in cuprous oxide. Thus, the defect concentration in the film layer is reduced, and the corrosion resistance of the film is improved. Moreover, as shown in the anodic polarization curves of the 70/30 samples after the 10-day erosion–corrosion test in Figure 5c, a low current density plateau is recorded next to the self-corrosion potential, so we state that the 70/30 copper–nickel alloy samples after the 10-day erosion–corrosion test show passivation behavior. In contrast, the 90/10 sample is not a typical passivation system because it does not have a passivation zone of low current density with a certain width.

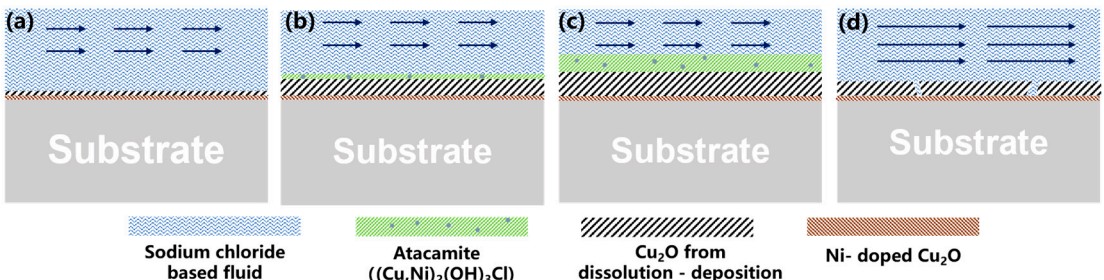

**Figure 12.** Effect of flow velocity on the corrosion product film of the 70/30 copper–nickel alloy in a sodium chloride-based fluid: (**a**–**c**) the schematics of the corrosion product film formation process at low flow velocities of 0.5–3 m/s; (**d**) the schematics of the corrosion product film formed at 4–4.7 m/s.

Additionally, nickel is the self-passive metal in sodium chloride solution. The addition of nickel can improve the corrosion resistance of copper alloys, as proved by A. M. Beccaria et al. [39]. The 70/30 copper–nickel alloy has a higher nickel content than the 90/10 copper–nickel alloy, so the 70/30 copper–nickel sample may have a better passivation capability. The EDX results in Figures 6 and 7 show that the ratio of Ni to Cu in the corrosion product film of the 70/30 copper–nickel sample is higher than that of the 90/10 copper–nickel sample, indicating that more nickel element is involved in the formation of the corrosion product film of the 70/30 copper–nickel sample, which is the result of the higher amount of Ni element in the 70/30 copper–nickel alloy than that in the 90/10 copper–nickel alloy.

Under the flow condition, the 90/10 and 70/30 copper–nickel alloys show the "critical flow velocity" phenomenon. From the results and discussion above, we can see that the optimum flow velocity for the formation of a protective corrosion product film in a 1 wt% NaCl solution is ~1.5 m/s for the 90/10 copper–nickel alloy and ~0.5 m/s for the 70/30 copper–nickel alloy. Based on whether there is a continuous corrosion product film as the judgment criterion, the "critical flow velocity" of the 90/10 copper–nickel alloy is between 3 m/s and 4 m/s, and the "critical flow velocity" of the 70/30 copper–nickel alloy is beyond 4.7 m/s.

When the corrosion product film is compact and has good protection for the substrate, as in the case of the 70/30 copper–nickel alloy at a flow velocity of 0.5–3 m/s, with the increase in flow velocity the anode process is promoted and the corrosion potential shifts negatively, indicating that the corrosion rate is controlled mainly by the anode process. The increase in flow velocity promotes the dissolution of cuprous ions from the outmost $Cu_2O$ lattice and the generation of cation vacancies ($V'_{Cu}(ox)$) (Reaction 9), and their diffusion towards the film/substrate interface, and their reaction with metallic Cu atoms (Reaction 10), thus promoting the anode reaction. When the corrosion product film is porous and dissolved oxygen can penetrate it, as in the case of the 90/10 copper–nickel alloy at a flow velocity of 3–4.7 m/s, with the increase in flow velocity the cathodic process is promoted

and the corrosion potential shifts passively, as shown in Figures 5a and 9, indicating that the corrosion rate is controlled mainly by the cathodic process. The increase in flow velocity promotes the mass transfer of dissolved oxygen diffusing through both the diffusion boundary layer and the corrosion product film, thus promoting the cathodic reaction.

### 5. Conclusions

(1) The corrosion product film on copper–nickel samples shows different corrosion resistance at different flow velocities. It is found that the corrosion product film formed at 1.5 m/s for the 90/10 copper–nickel tube and at 0.5 m/s for the 70/30 copper–nickel tube shows the best corrosion resistance compared with that formed at other flow velocities. A critical flow velocity phenomenon is found in both 90/10 and 70/30 copper–nickel samples. Based on whether there is a continuous corrosion product film as the judgment criterion, the "critical flow velocity" of the 90/10 copper–nickel alloy is between 3 m/s and 4 m/s, and the "critical flow velocity" of the 70/30 copper–nickel alloy is beyond 4.7 m/s;

(2) There is an atypical passivation region in the anodic polarization curve of the 90/10 copper–nickel alloy because the corrosion product film is not a typical passive film. By comparison of the polarization curves of the freshly polished 90/10 copper–nickel samples with those of the samples after the 10-day erosion–corrosion test, we can see that the presence of corrosion product film mainly inhibits the cathodic process, i.e., the oxygen reduction reaction (ORR) on the sample surface. The cathodic polarization curves will intersect the anodic polarization curves in different regions with the presence or absence of a corrosion product film on the substrate;

(3) The 70/30 copper–nickel alloy samples after the 10-day erosion–corrosion test show spontaneous passivation behavior, and typical passivation regions are recorded in their anodic curves, indicating that the corrosion product film on the 70/30 copper–nickel alloy is more similar to the passive film compared to that on the 90/10 copper–nickel alloy. By comparison of the polarization curves of the freshly polished 70/30 samples with those of the samples after the 10-day erosion–corrosion test, we can see that the presence of corrosion product film inhibits both anodic and cathodic processes of the 70/30 copper–nickel alloy, but at different flow velocities and with different surface films formed, the cathodic curve intersects the anodic curve in different regions.

**Author Contributions:** L.W.: Investigation, Methodology, Formal analysis, Writing—original draft. A.M.: Conceptualization, Methodology, Writing—review & editing. L.Z.: Investigation, Resources, Writing—review & editing. G.L., L.H. and Z.W.: Writing—review & editing. Y.Z.: Supervision, Writing—review & editing. All authors have read and agreed to the published version of the manuscript.

**Funding:** This work was supported by the National Natural Science Foundations of China (Grant number: 52171087 and 51871226).

**Institutional Review Board Statement:** Not applicable.

**Informed Consent Statement:** Not applicable.

**Data Availability Statement:** The raw and the processed data are available from A.M. upon reasonable request.

**Conflicts of Interest:** The authors declare no conflict of interest.

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
