# Peer review of "Erosion–Corrosion Behavior of 90/10 and 70/30 Copper–Nickel Tubes in 1 wt% NaCl Solution"

_metals, doi:10.3390/met13020401_

Round 1

Reviewer 1 Report

in an interesting paper to find the critical flow velocity. they use electrochemical techniques, supported by microscopy images and xps analysis, it is an excellent work, however i have the following comments

The document contains 36 times this sentence: "Error! Reference source not found".In the methods and materials section it is stated "which was detected by MFE-4 optical microscope and clearly shows equi axed grains and twins". What is not clear is which of the two alloys has grains and which has twins?

in the erosion corrosion test, it is necessary to know the change in mass, is it possible to know this?

In figure 9, a schematic mechanism of corrosion process is shown, it has the axes framed in potential vs. current, however it is very wide, so it would be appropriate to place ranges in which this behaviour can be established.

Reviewer 2 Report

I have reviewed this manuscript and I have the following comments:

1.  The text sould be revised the term Error! Reference source not found is found along the paper.

2. Correct the terms "In-suit" and "Ex-suit" in sections 3.1 and 3.2.

3. In the second paragraph, section 3.1.1, which is the meaning of "better density of the corrosion product ...".

4.. The terms Figure 1 and Figure 2 are inverted.

5. The authors should explain in more detail which is the meaning of the parameters (EC) obtained after the fitting of the EIS diagrams. The analysis is poor.

6. In page 12, the description of film resistance parameter is confused, should be rewritten.

7. In the section 3.1.2 Potentiodynamic polarization curves. The authors explain the protection of the 90/10 copper-nickel alloy as a atypical pasivation and for the 70/30 copper-nikel alloy with a typical passivation. However, both alloys are partially passivated and should not be compared with other metals or stainless steel. The authors should explain the semi-passive character of the films formed on both 90/10 and 70/30 copper-nickel alloys taking into account EDX profiles and XPS results. In fact the differences in the chemical composition of these films are related with the protection and the semiconductor properties which are related with the oxygen reduction reaction in the 1 wt.% NaCl solution.

8. Thus the discussion of the results should be improved. 

9. In page 22, the reaction 9, occurs at the film-solution interface. It is not clear in the text.

10. In section 4.2, first paragraph, which is the meaning of "spontaneous passivation"?

11. In page 25, at the end, How the authros explain that ionic and electronic conductivity decrease by the incorportation of Nickel into Cu2O film?

Reviewer 3 Report

In this work the authors have studied the corrosion behavior of two Cu-xNi alloy tubes (x=10 and 30 wt. %) in flowing 1 wt. % NaCl solution. Electrochemical methods have been used. It is found that the 70:30 alloy (Cu:Ni) provides a better corrosion performance because of passivating Ni. The paper is interesting and very detailed. It provides a plenty of results with several different flowing rates. A detailed discussion is also provided. The paper is worthy of publishing. Before publication, the following comments should be considered:

1.It would be helpful to provide a scheme/photograph of the electrochemical cell used to monitor the experiment in flowing water.

2.The choice of the electrolyte concentration should be explained (1 wt. % NaCl). Most experiments in literature are usually conducted in 3.5 wt. % NaCl.

3.The suggested corrosion products (Equations 1-7) should be confirmed by XRD data.

Round 2

Reviewer 3 Report

Authors answered my comments. The paper can be accepted for publication.